# Evaluation of Adjuvant Treatments for Adenoid Cystic Carcinoma of the Breast: A Population-Based, Propensity Score Matched Cohort Study from the SEER Database

**DOI:** 10.3390/diagnostics12071760

**Published:** 2022-07-21

**Authors:** Liu Yang, Chaobin Wang, Miao Liu, Shu Wang

**Affiliations:** Department of Breast Disease, Peking University People’s Hospital, Beijing 100044, China; yangliupkuph@sina.com (L.Y.); hzwcb1990@163.com (C.W.)

**Keywords:** breast cancer, adenoid cystic carcinoma, adjuvant, chemotherapy, radiation therapy, survival, prognosis

## Abstract

Adenoid cystic carcinoma (ACC) is an extremely rare type of breast cancer. The role of adjuvant treatments for ACC remains controversial. Patients with a histology-confirmed diagnosis of ACC of the breast were identified based on the SEER (Surveillance, Epidemiology and End Results) database. Propensity score matching (PSM) was performed to balance the baseline characteristics. The Kaplan–Meier method and Cox regression models were performed to determine the impact of the adjuvant chemotherapy (CT) and radiotherapy (RT) associated with breast cancer-specific survival (BCSS) and overall survival (OS). A total of 1036 patients with ACC of the breast were included. After a median follow-up of 11.3 years, the 10-year OS and BCSS rate was 76.2% and 92.6%, respectively. After PSM, adjuvant CT converted into neither OS (Log-rank *p* = 1.000) nor BCSS (Log-rank *p* = 0.900) benefits in patients with ACC of the breast. Adjuvant RT also did not improve OS (Log-rank *p* = 0.060) and BCSS (Log-rank *p* = 0.400). According to the univariate stratified analysis, adjuvant RT was favorable for OS in patients who underwent breast-conserving surgery (HR 0.66, 95% CI 0.45, 0.99, *p* = 0.042). From the multivariate analysis, histology grade and nodal status were independent prognostic factors for both OS and BCSS. In conclusion, ACC of the breast presented a favorable prognosis. Adjuvant treatment, especially adjuvant CT, might not be essential for patients with this disease.

## 1. Introduction

Breast cancer is a type of tumor with dramatic heterogeneity, which leads to differences in treatment strategies, including systemic therapy [1]. Even in patients with identical molecular subtypes as well as stages, systemic therapy options vary widely due to histological types of breast cancer [2]. Therefore, oncologists often consider histological types when formulating individualized systemic treatment strategies, especially for special and rare types of breast cancer [3].

Adenoid cystic carcinoma (ACC) of the breast is a particularly rare subtype of breast cancer, accounting for approximately 0.1–1% of all breast tumors [4,5]. ACC typically demonstrates the fusion of the transcription factor oncogene MYB to NFIB. In the rare breast ACC cases without the MYB-NFIB gene fusion, additional genomic alterations have been detected, including MYB gene amplification and MYBL1 gene rearrangement [6]. Although the immunohistochemical phenotype of most ACC of the breast is triple-negative, the prognosis is better than that of other triple-negative breast cancer [7].

Because of the rarity of this disease, there is no consensus on the optimal adjuvant therapy, including chemotherapy (CT) and radiotherapy (RT) [8,9]. Thus, we conducted this population-based and propensity score matched cohort study from the SEER (Surveillance, Epidemiology and End Results) database to assess the impact of adjuvant treatments in patients with ACC of the breast.

## 2. Materials and Methods

### 2.1. Patients

Patients with a histology-confirmed diagnosis of ACC of the breast (ICD-O-3 morphology codes: 8200/3) were identified based on the SEER database from 1975 to 2019. Patients included were required to meet the following criteria: (1) at stage T1-4N0-3M0 according to the 6th edition of the AJCC/UICC (American Joint Committee on Cancer/Union for International Cancer Control); (2) with complete records on the live status and survival months. To compare survival between ACC and other histological types with a favorable prognosis, patients diagnosed with tubular carcinoma (TC, ICD-O-3 morphology codes: 8211/3) and mucinous carcinoma (MC, ICD-O-3 morphology codes: 8480/3) were also included in this study.

### 2.2. Clinicopathological Factors

Clinicopathological factors of included patients were extracted from the SEER database, including gender, age, race, year of diagnosis, histology grade, tumor stage, nodal status, TNM stage as well as the expression of the estrogen receptor (ER), progesterone receptor (PR) and human epidermal growth factor receptor 2 (Her2). ER-positive and/or PR-positive patients were then categorized as hormone receptor (HR) positive, and otherwise HR negative. Breast surgery was divided into breast conserving surgery (BCS) and total mastectomy (MAST). The records on adjuvant treatments, including chemotherapy (CT) and radiotherapy (RT), were also obtained from the SEER database.

### 2.3. Survival Outcomes

The SEER database provided information on the live status and survival months. The survival outcomes included overall survival (OS) and breast cancer-specific survival (BCSS). The length of OS was calculated from the date breast cancer was diagnosed to the date of death for any reason, and BCSS was calculated from the date breast cancer was diagnosed to the date of death for reasons related to breast cancer.

### 2.4. Propensity Score Matching (PSM)

When comparing survival between patients with ACC of the breast according to different adjuvant treatment strategies (with CT vs. without CT; with RT vs. without RT), propensity score matching was used to balance the baseline characteristics. We performed a 1:1 nearest-neighbor matching procedure within a caliper of 0.02 and all clinic and pathological factors were included in the matching model. The balance between the two groups before and after matching was assessed using standardized mean differences (SMD) and *p*-value by Chi-square test or *t*-test. SMD > 0.20 or *p*-value < 0.05 were considered imbalanced.

### 2.5. Statistical Analysis

Continuous variables were reported as mean and standard deviation, whereas categorical variables were reported as percentages. Statistical differences in the distribution of continuous and categorical variables were conducted by *t*-test and chi-square test respectively.

Survival analysis was performed by the Kaplan–Meier method; thus, median survival time and 5-year and 10-year survival rates were estimated. Survival differences between cohorts were compared by Log-rank test. Univariate and multivariate Cox proportional hazards regression analyses were used to select independent prognostic factors influencing OS and BCSS in patients with ACC of the breast. Factors that showed a univariate connection with survival or were considered clinically relevant were entered into the multivariate COX proportional-hazard regression model. Interaction terms, which were tested by qualitative method and univariate stratified COX proportional-hazard regression model, were used to investigate whether the association between adjuvant treatment and survival outcomes differed according to certain clinicopathological factors. Two-tailed *p*-values < 0.05 were considered statistically significant. All analyses were conducted using R software (Beijing, China, http://www.Rproject.org accessed on 15 May 2022).

## 3. Results

Excluding 8 patients with de novo stage IV, 1036 out of 1044 patients with ACC of the breast were included in the analyses. The flow diagram of the process of patients’ selection and analyses is presented in Figure 1.

### 3.1. Patients Characteristics

Among all included patients, the mean age at diagnosis of ACC of the breast was 62 years (standard deviation: 13.13 years), and 41.9% (434/1036) of them were diagnosed in recent years (2010–2019). Patients with ACC of the breast were more likely to present low-risk clinicopathological features: 46.7% (484/1036) of them with histology grade I–II tumors; 61.9% (641/1036) at nodal stage N0; 64.1% (664/1036) at TNM stage I–II. More than half of the patients (58.6%, 607/1036) were ER negative. Because the SEER database started collecting information on Her2 status since 2010, data on her2 expression were missing in 639 patients. Of 397 patients with known Her2 status, 390 (98.2%) were Her2 negative. In terms of adjuvant treatments, 99 (9.6%, 99/1036) patients received adjuvant CT and 422 (40.7%, 422/1036) underwent adjuvant RT. Clinicopathological characteristics of included patients with ACC of the breast stratified by adjuvant treatments are listed in Table 1.

### 3.2. The Impact of Adjuvant CT on Survival in Patients with ACC of the Breast

After a median follow-up of 11.3 years (95% CI 10.3, 12.2), 357 (34.5%, 357/1036) patients had died for any reason and 77 (7.4%, 77/1036) for reasons related to breast cancer. There was no statistically significant difference in OS between patients with ACC of the breast who received adjuvant CT (N = 99) and those who did not (N = 937) (Log-rank *p* = 0.800, Figure 2a). Patients who received adjuvant CT presented a worse BCSS (Log-rank *p* < 0.001, Figure 2c) compared with those without adjuvant CT. After PSM, 41 matched patients were in both cohorts. Clinicopathological characteristics of included patients with ACC of the breast stratified by adjuvant CT before and after PSM are listed in Appendix A. However, adjuvant CT converted into neither OS (Log-rank *p* = 1.000, Figure 2b) nor BCSS (Log-rank *p* = 0.900, Figure 2d) benefits in patients with ACC of the breast after the matching procedure.

### 3.3. The Impact of Adjuvant RT on Survival in Patients with ACC of the Breast

The median OS was 19.1 years (95% CI 17.1, 24.1) among patients who received adjuvant RT (N = 422) compared with 19.8 years (95% CI 17.7, 22.0) among patients who did not (N = 614). Adjuvant RT converted into OS benefit in patients with ACC of the breast (Log-rank *p* = 0.040, Figure 2e) but not into BCSS benefit (Log-rank *p* = 0.100, Figure 2g). After PSM, 79 matched patients were in both cohorts. Clinicopathological characteristics of included patients with ACC of the breast stratified by adjuvant RT before and after PSM are listed in Appendix A. There was no statistically significant difference in both OS (Log-rank *p* = 0.060, Figure 2f) and BCSS (Log-rank *p* = 0.400, Figure 2h) between patients with and without adjuvant RT after the matching procedure.

### 3.4. Univariate Stratified Analysis of the Impact of Adjuvant Treatments on Survival in Patients with ACC of the Breast

Included patients with ACC of the breast were then stratified into subgroups according to age at diagnosis, histology grade, hormone receptor status, nodal status and TNM stage to further explore the prognostic role of adjuvant CT. The prognostic value of adjuvant CT was not favorable for both OS and BCSS among all subgroups of patients (shown in Figure 3a,b). Patients were also stratified by nodal status and breast surgery methods to observe the relationship between adjuvant RT and survival. As shown in Figure 3c, the prognostic value of adjuvant RT was consistently favorable for OS in the following subgroups of patients: underwent BCS (stratified HR 0.66, 95% CI 0.45, 0.99, stratified *p* = 0.042, *p* for interaction = 0.228), node negative (stratified HR 0.71, 95% CI 0.52, 0.95, stratified *p* = 0.023, *p* for interaction = 0.001) and node negative with BCS (stratified HR 0.62, 95% CI 0.40, 0.95, stratified *p* = 0.028, *p* for interaction = 0.594). However, the positive prognostic value of adjuvant RT for BCSS was only observed in node-negative patients (stratified HR 0.46, 95% CI 0.23, 0.92, stratified *p* = 0.028, *p* for interaction = 0.379) but not in patients who underwent BCS (shown in Figure 3d).

### 3.5. Univariate and Multivariate Analysis of Clinicopathological Factors Influencing the Survival of Patients with ACC of the Breast

A total of 399 patients with complete data records on clinicopathological information (apart from Her2 status) were included in univariate and multivariate analysis. The details about univariate and multivariate analysis are listed in Table 2.

The following factors influenced both OS and BCSS independently: histology grade III–IV and nodal status (negative vs. positive). In addition, age at diagnosis (≤60 vs. >60 years) was an independent prognostic factor for OS. Adjuvant CT and RT did not decrease the hazard of mortality of patients with ACC of the breast on the multivariate models.

### 3.6. Comparison of Survival between Patients Diagnosed with ACC, MC and TC of the Breast

As shown in Figure 1, 12,231 patients diagnosed with TC and 27,878 patients diagnosed with MC were also included in this study to make a comparison of survival with patients diagnosed with ACC of the breast (N = 1036). The 5-year and 10-year OS and BCSS rates of patients with different histological types of breast cancer are listed in Table 3.

The median OS was 19.8 (95% CI 18.2, 22.0), 14.2 (95% CI 14.0, 14.5) and 21.5 (95% CI 21.1, 22.1) years among patients diagnosed with ACC, MC and TC, respectively (Log-rank *p* < 0.001, Figure 4a). There was no statistically significant difference in BCSS between patients diagnosed with ACC and MC of the breast (Log-rank *p* = 0.380, Figure 4b).

## 4. Discussion

This population-based cohort study from the SEER database showed that patients with ACC of the breast could not benefit from adjuvant CT and RT after the matching procedure. According to multivariate analysis, neither adjuvant RT nor CT decreased the hazard of mortality of patients with ACC of the breast.

ACC of the breast was a rare histological type of adenocarcinoma and presented an almost triple-negative molecular subtype [10,11]. Consistent with the current study, 58.6% (607/1036) of patients with ACC of the breast in our study were ER negative. Among patients with Her2 expression data, 98.2% (390/397) were Her2 negative. Interestingly, quite different from triple-negative breast cancer, we observed that patients with ACC of the breast had a favorable prognosis with 5-year and 10-year BCSS rates of 95.3% and 92.6%, respectively. In addition, no statistically significant difference existed in OS between ACC and MC, which was favorable histology and generally ER positive and Her2 negative [12,13]. Given its rarity and good prognosis, the value of adjuvant therapy in patients with ACC of the breast remained controversial and led to large variations in clinical practice [8,14].

Adjuvant CT was provided less often for patients with ACC of the breast. Among 1036 patients in the SEER database, only 99 (9.6%) received adjuvant CT. Similarly, a published study from NCDB (National Cancer Database) reported that 11.3% of patients with ACC of the breast received adjuvant CT [15], which reflected a negative attitude of physicians in choosing adjuvant CT for patients with ACC of the breast. This might be explained by several reasons. First, it was a fact that the vast majority of patients with ACC of the breast were clinically low risk: 46.7% (484/1036) were histology grade I to II and 61.9% (641/1036) were node negative. Second, distant relapses were rarely seen: in the Rare Cancer Network (RCN) study, in which 61 patients were enrolled, only 4 (6.6%, 4/61) had distant failures after a median follow-up of 79 months. This seems to suggest that adjuvant CT was not necessary and cost-effective for patients with ACC of the breast.

However, for patients at clinical high risk, such as histology Grade III–IV and positive nodal status, our study found that the hazard of mortality increased. Whether adjuvant CT could optimize the prognosis of patients at clinical high-risk was still a concern. NCCN (National Comprehensive Cancer Network) guidelines recommended adjuvant CT only for node positive patients with ACC of the breast even though very few data have been published yet [16]. In our study, patients with adjuvant CT were associated with younger age, higher histology grade, larger tumor size, nodal involved and ER negative. As a possible result, the BCSS of patients with adjuvant CT was even worse than that of patients without adjuvant CT. After balancing clinicalopathological factors, adjuvant CT still failed to improve either OS or BCSS of matched patients. Consistent results were also observed in univariate subgroup analyses. Nonetheless, a long-term follow-up for such patients is warranted due to the survival data were not yet mature.

Among 1036 patients in the SEER database, 422 received adjuvant RT: 346 after BCS and 16 with positive nodal status. Because the incidence of nodal metastasis (2.6%, 27/1036) was very low, it was not possible to analyze the association between adjuvant RT and nodal status. The role of adjuvant RT in locoregional control and survival after BCS in ACC of the breast remained unclear. The RCN study reported that adjuvant RT after BCS significantly correlated with locoregional control rate in patients with ACC of the breast [17]. However, survival was not influenced by the use of postoperative RT. Our study showed that adjuvant RT improved OS but not BCSS, which was consistent with another analysis based on the SEER database conducted by Sun et al. [18]. Furthermore, the results of the univariate stratified analysis suggested that adjuvant RT was still favorable for OS in patients who underwent BCS. However, the prognostic value of adjuvant RT was not statistically significant after PSM. Despite the lack of convincing evidence, the current guidelines recommended that patients with ACC of the breast should receive adjuvant RT routinely after BCS.

In this study, 16.3% (169/1036) of patients with ACC of the breast were ER positive. The previous studies reported that the positive rate of ER in patients with this disease was about 15–18%, which was consistent with the result of our study [15,18]. There was no difference in OS (Log-rank *p* = 0.844) and BCSS (Log-rank *p* = 0.382) between ER-positive and ER-negative patients with ACC of the breast. According to the multivariate analysis, HR status was not an independent prognostic factor for patients with ACC of the breast. Only a few patients have received endocrine therapy in the published studies [15,17]. Therefore, the effect of endocrine therapy on ER-positive patients with ACC of the breast was also unclear. It is a pity that there has been no record of adjuvant endocrine therapy in the SEER database. More data need to report in the future.

Due to the extremely low incidence of ACC of the breast, it was not feasible to design and conduct prospective trials to assess the effect of adjuvant therapy on this rare type of breast cancer. Thus, we conducted this matched cohort study in a large population-based setting from the SEER database. Our study also had some shortcomings. The first was the nature of retrospective studies, which caused inherent biases in this study. The second was limitations in the SEER database, for example, missing data on Her2 expression.

In conclusion, ACC is a rare histological type of breast cancer with a favorable prognosis and adjuvant treatment, especially adjuvant CT, might not be essential for these patients.

## Figures and Tables

**Figure 1 diagnostics-12-01760-f001:**
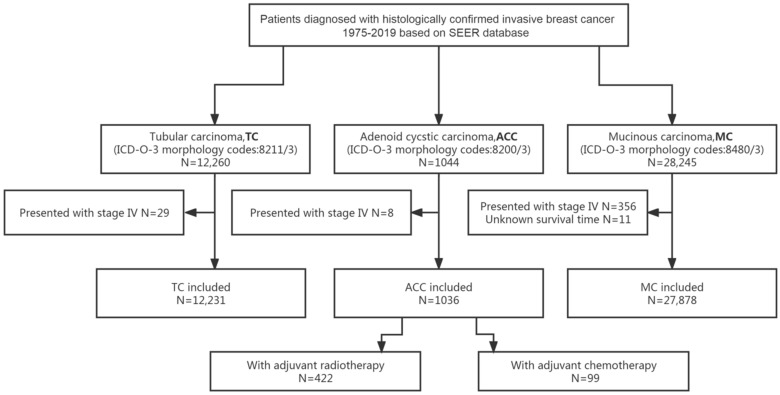
Flow diagram for the selection of the study cohort. Abbreviations: ACC = adenoid cystic carcinoma of the breast; ICD-O-3 = International Classification of Diseases for Oncology, third edition; MC = mucinous carcinoma of the breast; SEER = Surveillance, Epidemiology and End Results; TC = tubular carcinoma of the breast.

**Figure 2 diagnostics-12-01760-f002:**
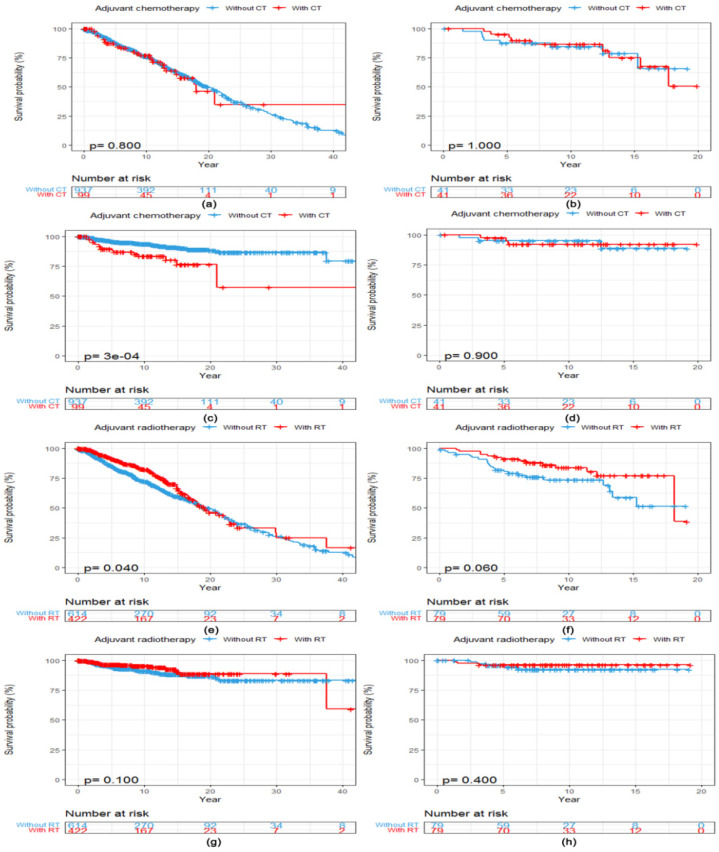
Kaplan–Meier curves showing a comparison of survival among patients with ACC of the breast according to adjuvant treatments. OS curves of patients stratified by adjuvant CT (**a**) before and (**b**) after PSM. BCSS curves of patients stratified by adjuvant CT (**c**) before and (**d**) after PSM. OS curves of patients stratified by adjuvant RT (**e**) before and (**f**) after PSM. BCSS curves of patients stratified by adjuvant RT (**g**) before and (**h**) after PSM. Abbreviations: ACC = adenoid cystic carcinoma of the breast; BCSS = breast cancer-specific survival; CT = chemotherapy; OS = overall survival; PSM = propensity score matching; RT = radiotherapy.

**Figure 3 diagnostics-12-01760-f003:**
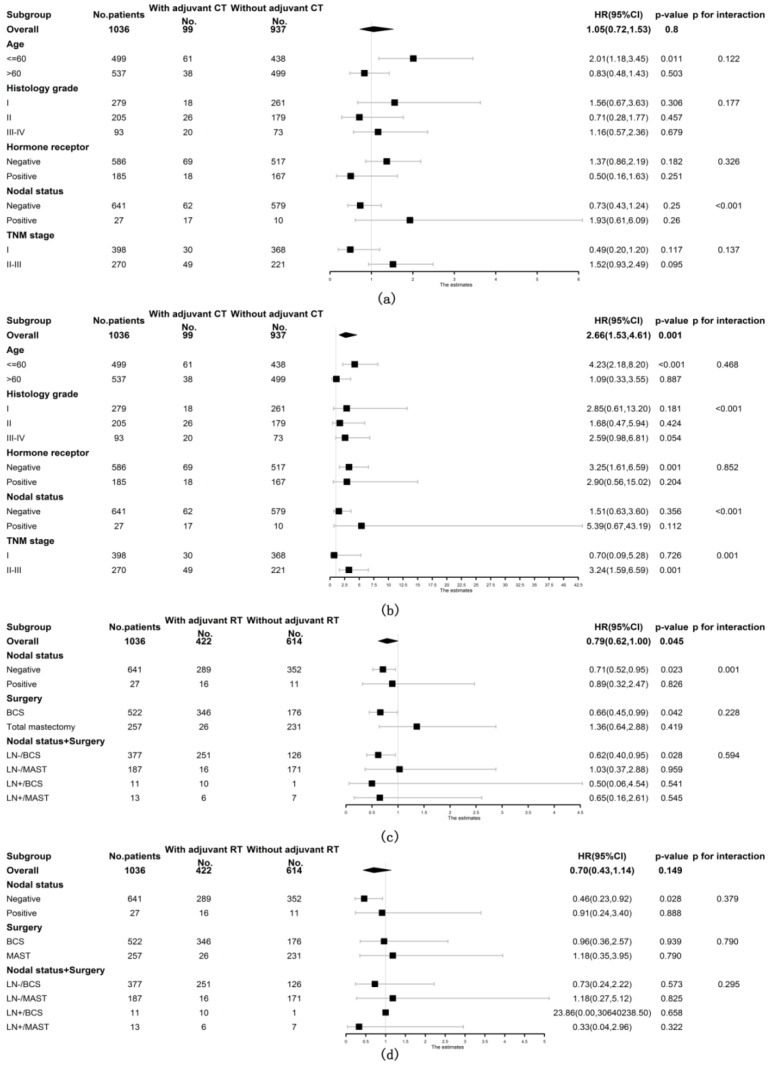
Forest plots of interaction and univariate subgroup analyses on the association between adjuvant treatments and survival of patients with ACC of the breast. The association between adjuvant chemotherapy and (**a**) OS and (**b**) BCSS of patients with ACC of the breast. The association between adjuvant radiotherapy and (**c**) OS and (**d**) BCSS of patients with ACC of the breast. Abbreviations: ACC = adenoid cystic carcinoma of the breast; BCS = breast conserving surgery; BCSS = breast cancer-specific survival; CT = chemotherapy; HR = hazard risk; LN- = lymph node negative; LN+ = lymph node positive; MAST = mastectomy; No. = number of; OS = overall survival; RT = radiotherapy; 95% CI = 95% confidence interval.

**Figure 4 diagnostics-12-01760-f004:**
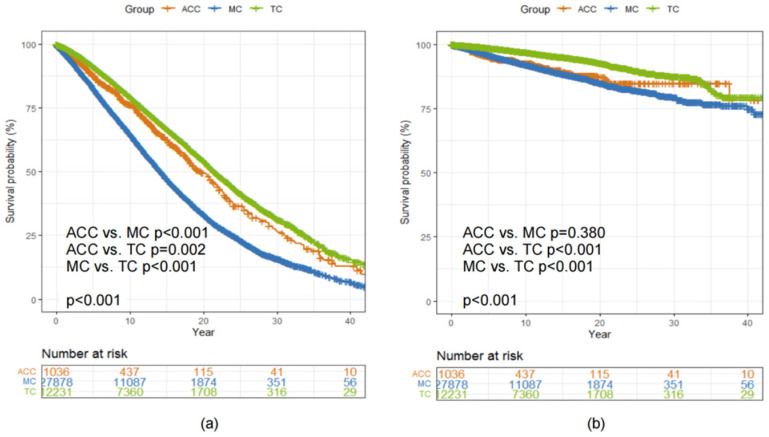
Kaplan–Meier curves showing a comparison of (**a**) OS and (**b**) BCSS among patients with ACC, MC and TC of the breast. Abbreviations: ACC = adenoid cystic carcinoma of the breast; BCSS = breast cancer-specific survival; MC = mucinous carcinoma of the breast; OS = overall survival; TC = tubular carcinoma of the breast.

**Table 1 diagnostics-12-01760-t001:** Clinic and pathological characteristics of patients with ACC of the breast (N = 1036) stratified by adjuvant treatments.

Characteristic	Overall N = 1036	Adjuvant CT		*p*-Value	Adjuvant RT		*p*-Value
		Without N = 937	With N = 99		Without N = 614	With N = 422	
Gender (%)							
Female	1024 (98.8)	926 (98.8)	98 (99.0)	1.000	607 (98.9)	417 (98.8)	1.000
Male	12 (1.2)	11 (1.2)	1 (1.0)		7 (1.1)	5 (1.2)	
Age (mean (SD))	62 (13.13)	62 (13.15)	56 (11.51)	<0.001	63 (13.88)	60 (11.75)	<0.001
Race (%)							
White	861 (83.1)	783 (83.6)	78 (78.8)	0.354	508 (82.7)	353 (83.6)	0.118
Black	88 (8.5)	75 (8.0)	13 (13.1)		47 (7.7)	41 (9.7)	
Other *	80 (7.7)	73 (7.8)	7 (7.1)		56 (9.1)	24 (5.7)	
UNK	7 (0.7)	6 (0.6)	1 (1.0)		3 (0.5)	4 (0.9)	
Year of diagnosis (%)							
1975–1999	256 (24.7)	248 (26.5)	8 (8.1)	<0.001	201 (32.7)	55 (13.0)	<0.001
2000–2009	346 (33.4)	294 (31.4)	52 (52.5)		191 (31.1)	155 (36.7)	
2010–2019	434 (41.9)	395 (42.2)	39 (39.4)		222 (36.2)	212 (50.2)	
Histology Grade (%)							
I	279 (26.9)	261 (27.9)	18 (18.2)	<0.001	156 (25.4)	123 (29.1)	0.002
II	205 (19.8)	179 (19.1)	26 (26.3)		109 (17.8)	96 (22.7)	
III	79 (7.6)	62 (6.6)	17 (17.2)		39 (6.4)	40 (9.5)	
IV	14 (1.4)	11 (1.2)	3 (3.0)		7 (1.1)	7 (1.7)	
UNK	459 (44.3)	424 (45.3)	35 (35.4)		303 (49.3)	156 (37.0)	
Tumor stage (%)							
T1	406 (39.2)	373 (39.8)	33 (33.3)	<0.001	215 (35.0)	191 (45.3)	<0.001
T2–T4	262 (25.3)	216 (23.1)	46 (46.5)		148 (24.1)	114 (27.0)	
UNK	368 (35.5)	348 (37.1)	20 (20.2)		251 (40.9)	117 (7.7)	
Nodal status (%)							
Negative	641 (61.9)	579 (61.8)	62 (62.6)	<0.001	352 (57.3)	289 (68.5)	<0.001
Positive	27 (2.6)	10 (1.1)	17 (17.2)		11 (1.8)	16 (3.8)	
UNK	368 (35.5)	348 (37.1)	20 (20.2)		251 (40.9)	117 (27.7)	
TNM stage (%)							
I	398 (38.4)	368 (39.3)	30 (30.3)	<0.001	208 (33.9)	190 (45.0)	<0.001
II	266 (25.7)	220 (23.5)	46 (46.5)		152 (24.8)	114 (27.0)	
III	4 (0.4)	1 (0.1)	3 (3.0)		3 (0.5)	1 (0.2)	
UNK	368 (35.5)	348 (37.1)	20 (20.2)		251 (40.9)	117 (27.7)	
ER (%)							
Negative	607 (58.6)	537 (57.3)	70 (70.7)	0.006	314 (51.1)	293 (69.4)	<0.001
Positive	169 (16.3)	152 (16.2)	17 (17.2)		96 (15.6)	73 (17.3)	
UNK	260 (25.1)	248 (26.5)	12 (12.1)		204 (33.2)	56 (13.3)	
PR (%)							
Negative	677 (65.3)	606 (64.7)	71 (71.7)	0.001	355 (57.8)	322 (76.3)	<0.001
Positive	96 (9.3)	80 (8.5)	16 (16.2)		53 (8.6)	43 (10.2)	
UNK	263 (25.4)	251 (26.8)	12 (12.1)		206 (33.6)	57 (13.5)	
Her2 (%)							
Negative	390 (37.6)	353 (37.7)	37 (37.4)	0.229	192 (31.3)	198 (46.9)	<0.001
Positive	7 (0.7)	5 (0.5)	2 (2.0)		3 (0.5)	4 (0.9)	
UNK	639 (61.7)	579 (61.8)	60 (60.6)		419 (68.2)	220 (52.1)	
Surgery (%)							
BCS	522 (50.4)	469 (50.1)	53 (53.5)	<0.001	176 (28.7)	346 (82.0)	<0.001
Total mastectomy	257 (24.8)	218 (23.3)	39 (39.4)		231 (37.6)	26 (6.2)	
UNK	257 (24.8)	250 (26.7)	7 (7.1)		207 (33.7)	50 (11.8)	

* Other: American Indian/AK native, Asian/Pacific Islander. Abbreviations: BCS = breast conserving surgery; CT = chemotherapy; ER = estrogen receptor; Her2 = human epidermal growth factor receptor 2; PR = progesterone receptor; RT = radiotherapy; SD = standard deviation; TNM = tumor-node-metastasis; UNK = Unknown.

**Table 2 diagnostics-12-01760-t002:** Prognostic factors influencing the survival of patients with ACC of the breast (N = 399).

Characteristic	No.	OS				BCSS			
		Univariate		Multivariate		Univariate		Multivariate	
		HR (95% CI)	*p*-Value	HR (95% CI)	*p*-Value	HR (95% CI)	*p*-Value	HR (95% CI)	*p*-Value
Age									
≤60	207	3.88 (2.49, 6.04)	<0.001	3.90 (2.48, 6.12)	<0.001	1.05 (0.51, 2.14)	0.904	1.10 (0.52, 2.33)	0.810
>60	192								
Year of diagnosis									
1975–2009	216	0.69 (0.43, 1.12)	0.135	0.81 (0.50, 1.33)	0.394	0.36 (0.15, 0.90)	0.029	0.55 (0.21, 1.43)	0.217
2010–2019	183								
Histology Grade									
I	186	Ref.		Ref.		Ref.		Ref.	
II	146	0.94 (0.60, 1.49)	0.803	0.89 (0.55, 1.43)	0.625	1.47 (0.53, 4.04)	0.460	0.89 (0.30, 2.60)	0.827
III–IV	67	2.20 (1.37, 3.52)	0.001	1.91 (1.17, 3.12)	0.010	7.06 (2.88, 17.34)	<0.001	5.22 (2.02, 13.49)	0.001
Tumor stage									
T1	231	1.31 (0.90, 1.92)	0.173	1.72 (0.41, 7.21)	0.461	3.56 (1.63, 7.77)	0.001	2.20 (0.22, 21.98)	0.501
T2–T4	168								
Nodal status									
Negative	380	4.15 (2.27, 7.61)	<0.001	4.85 (2.09, 11.24)	<0.001	11.85 (5.40, 26.00)	<0.001	5.73 (2.02, 16.21)	0.001
Positive	19								
TNM stage									
I	227	1.43 (0.98, 2.10)	0.066	0.75 (0.17, 3.31)	0.699	4.08 (1.82, 9.17)	0.001	1.05 (0.09, 12.25)	0.971
II-III	172								
Hormone receptor *									
Negative	316	0.81 (0.48, 1.36)	0.422	0.82 (0.47, 1.45)	0.496	0.27 (0.07, 1.15)	0.077	0.27 (0.06, 1.29)	0.100
Positive	83								
Surgery									
BCS	270	1.09 (0.73, 1.63)	0.672	0.74 (0.45, 12.4)	0.258	2.41 (1.18, 4.93)	0.016	1.78 (0.66, 4.81)	0.254
Total mastectomy	129								
Adjuvant RT									
Without	202	0.73 (0.49, 1.07)	0.106	0.64 (0.39, 1.05)	0.080	0.64 (0.31, 1.33)	0.233	0.65 (0.24, 1.77)	0.400
With	197								
Adjuvant CT									
Without	345	1.14 (0.68, 1.92)	0.625	0.91 (0.49, 1.70)	0.764	3.17 (1.48, 6.78)	0.003	1.69 (0.64, 4.48)	0.294
With	54								

* ER positive and/or PR positive was categorized as hormone receptor positive. ER negative and PR negative was categorized as hormone receptor negative. Abbreviations: BCS = breast conserving surgery; BCSS = breast cancer-specific survival; CT = chemotherapy; HR = hazard rate; No. = number of patients; OS = overall survival; PSM = propensity scores matching; Ref. = reference; RT = radiotherapy; SMD = standardized mean difference; TNM = tumor-node-metastasis; 95% CI = 95% confidence interval.

**Table 3 diagnostics-12-01760-t003:** Survival (OS and BCSS) rates in patients with ACC, TC and MC of the breast from the SEER database between 1975 and 2019.

Survival Rate	ACC N = 1036	TC N = 12,231	MC N = 27,878
5-year OS% (95% CI)	88.09 (85.99, 90.20)	91.30 (90.80, 91.80)	82.90 (82.43, 83.37)
10-year OS% (95% CI)	76.24 (73.25, 79.30)	79.30 (78.56, 80.10)	64.79 (64.14, 65.43)
5-year BCSS% (95% CI)	95.30 (93.90, 96.70)	98.60 (98.40, 98.80)	96.00 (95.70, 96.20)
10-year BCSS% (95% CI)	92.60 (90.70, 94.50)	96.90 (96.60, 97.20)	92.10 (91.70, 92.50)

Abbreviations: ACC = adenoid cystic carcinoma of the breast; BCSS = breast cancer-specific survival; MC = mucinous carcinoma of the breast; OS = overall survival; SEER = Surveillance, Epidemiology and End Results; TC = tubular carcinoma of the breast; 95% CI = 95% confidence interval.

## Data Availability

All data generated or analyzed during the study are included in the published paper.

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
