# Peer review of "Evaluation of Adjuvant Treatments for Adenoid Cystic Carcinoma of the Breast: A Population-Based, Propensity Score Matched Cohort Study from the SEER Database"

_diagnostics, 2022, doi:10.3390/diagnostics12071760_

Round 1

Reviewer 1 Report

This is a large retrospektive study on Adenoidcystic carcinoma of the breast based on the SEER database. The study confirms the observations of other much smaller series that ACC is a triple negative tumor of good prognosis and chemotherapy does not influence the outcome; and also that histology grade (subtyping ACC) and node status are the most mportant prognostic factors.

Minor corrections are needed. 

- The proportion of ER positive cases is rather high, it needs a comment in the Discussion.

-The start search year in SEER is stated as 1975 in the text and 1979 in figure 1.

- I would skip words as extremely or surprisingly as the results confirm already published characteristics of ACC.

Author Response

Point 1: The proportion of ER positive cases is rather high, it needs a comment in the Discussion.

Response 1: Thanks very much for taking the time to review this manuscript. We appreciate all your comments and suggestions. In this study, 16.3% (169/1,036) of patients with ACC of the breast were ER-positive. The previous studies reported that the positive rate of ER in patients with this disease was about 15-18% which was consistent with the result of our study [1,2]. There was no difference in OS (Log-rank p=0.844) and BCSS (Log-rank p=0.382) between ER-positive and ER-negative patients with ACC of the breast. According to the multivariate analysis, HR status was not an independent prognostic factor for patients with ACC of the breast. Only a few patients have received endocrine therapy in the published studies [1,3]. Therefore the effect of endocrine therapy on ER-positive patients with ACC of the breast was also unclear. It is a pity that there has been no record of adjuvant endocrine therapy in the SEER database. More data need to report in the future. According to your suggestions, we have added the above comments to the discussion section of the manuscript (Line 376-386).

References

  1. Nandini Kulkarni, Christopher M. Pezzi, Jon M. Greif, DO, et al. Rare Breast Cancer: 933 Adenoid Cystic Carcinomas from the National Cancer Data Base. Ann Surg Oncol (2013) 20:2236–2241.
  2. Jia-Yuan Sun, San-Gang Wu, Shan-Yu Chen, et al. Adjuvant radiation therapy and survival for adenoid cystic carcinoma of the breast. Breast. 2017 Feb; 31:214-218.
  3. Kaouthar Khanfir,Adel Kallel,Sylviane Villette,et al. Management of adenoid cystic carcinoma of the breast: a Rare Cancer Network study. Int J Radiat Oncol Biol Phys. 2012 Apr 1;82(5):2118-24.

Point 2: The start search year in SEER is stated as 1975 in the text and 1979 in figure 1.

Response 2: Thanks very much for taking the time to review this manuscript. We appreciate all your comments and suggestions. We are very sorry to admit the error in Figure 1: the correct year is 1975, not 1979. We have revised the year in Figure 1 and Table 3 (the year in the title).

Point 3: I would skip words as extremely or surprisingly as the results confirm already published characteristics of ACC.

Response 3: Thanks very much for taking the time to review this manuscript. We agree with your suggestions and will further explore this field in future research. Most previous studies observed the characteristics of ACC of the breast based on a small sample of single-center population, so the strength of evidence may be limited. A few studies based on public databases have also analyzed the relationship between survival and adjuvant treatments in patients with ACC of the breast, but most have focused on adjuvant radiotherapy. Our study was based on the SEER database, which not only analyzed the impact of adjuvant radiotherapy on survival but also analyzed the prognostic value of adjuvant chemotherapy. Additionally, the method of stratified analysis was also used to explore the possible effect modifiers. In terms of statistical analysis, the PSM method was used to match the patients according to clinicopathological characteristics, to better control underlying confounding factors. We sincerely accept your opinions and will conduct further studies on breast cancer with rare histological types in the future, especially those that are controversial in terms of adjuvant therapy. Thank you again for your suggestions.

Reviewer 2 Report

Overall, it is an interesting article. Needs some clarifications and editing of the language:

line 52-patients identified in database from 1975-2019, in diagram Fig. 1 1979-2019

Language editing needed- to suggest a few changes: 

Line 12: SEER ((Surveillance, Epidemiology, and End Results, 12 (SEER) the words first

Line 23: ACC of the breast cancer

line 56-57: to compared, which generally recognized

line 103: included  into in the

line 129-130: patients underwent received adjuvant CT

line 155: patients had dead had died for any reason

line 158: patients with who received adjuvant CT

line 166-167: underwent who received adjuvant RT // while compared with 19.8 y among patients who did not

line 208: was consistently favorable

line 210-211, 2013: nodal node negative

line 215: but not in patients who underwent BCT

lines 314-354: multiple language refinements needed

line 375:a great loss of missing data  ..... lacks lack of records

line 377: ACC was is a rare

Author Response

Point 1: line 52-patients identified in database from 1975-2019, in diagram Fig. 1 1979-2019.

Response 1: Thanks very much for taking the time to review this manuscript. We appreciate all your comments and suggestions. We are very sorry to admit the error in Figure 1: the correct year is 1975, not 1979. We have revised the year in Figure 1 and Table 3 (the year in the title).

Point 2: Language editing needed to suggest a few changes.

Response 2: Thanks very much for taking the time to review this manuscript. We are very sorry for the language problems in the manuscript and have made careful corrections according to your suggestions (shown in the table below). Thank you again for your caring advice.

Location

Before Correction

After Correction

Line 12

base on SEER

based on the SEER

Line 12-13

((Surveillance, Epidemiology, and End Results, SEER)

((Surveillance, Epidemiology, and End Results, SEER)

Line 20

According to univariate stratified analysis

According to the univariate stratified analysis

Line 22

From multivariate analysis

From the multivariate analysis

Line 24

ACC of the breast cancer

ACC of the breast cancer

Line 25

for patient with this disease

for patients with this disease

Line 33-34

oncologists often take histological types into consideration

oncologists often take consider histological types into consideration

Line 42

is triple-negative subtype

is triple-negative subtype

Line 52

based on SEER database

based on the SEER database

Line 55

records on live status

records on the live status

Line 56

In order to compared survival

In order to To compared survival

Line 57

histological types which generally recognized with a favourable prognosis

histological types which generally recognized with a favourable prognosis

Line 61

extracted from SEER

extracted from the SEER

Line 65

ER-positive and/or PR-positive was then

ER-positive and/or PR-positive was were then

Line 69

from SEER database

from the SEER database

Line 71

on live status

on the live status

Line 81

in matching model

in the matching model

Line 81

Balance between the two groups

The balance between the two groups

Line 83

Chi square test

Chi-square test

Line 83

t test

t-test

Line 87

as percentage

as percentages

Line 91

survival rate were estimated

survival rates were estimated

Line 94

Factors showed

Factors that showed

Line 95

or considered clinically relevant

or were considered clinically relevant

Line 96

into multivariate COX

into the multivariate COX

Line 104

included into analyses

included into the analyses

Figure 1

1979-2019

1975-2019

Line 125

Patient with ACC of the breast

Patients with ACC of the breast

Line 128

half of patients

half of the patients

Line 131

99 patients underwent

99 patients underwent received

Line 134

treatments is listed

treatments is are listed

Line 158

patients had dead for any reason

patients had dead died for any reason

Line 161

Patients with adjuvant CT

Patients with who received adjuvant CT

Line 163-164

characteristics of ... is listed in Table S1

characteristics of ... is are listed in Table S1

Line 167

after matching procedure

after the matching procedure

Line 169

patients underwent adjuvant RT

patients underwent who received adjuvant RT

Line 170

while 19.8 years among patients did not

While compared with 19.8 years among patients who did not

Line 174-175

characteristics of ... is listed in Table S2

characteristics of ... is are listed in Table S2

Line 177

after matching procedure

after the matching procedure

Line 213

was consistent favorable

was consistently favorable

Line 215-216, 219

nodal negative

nodal node negative

Line 220

patients underwent BCS

patients who underwent BCS

Line 252

apart for Her2 status

apart for from Her2 status

Line 281

The following two factors

The following two factors

Line 283

Adjuvant CT, as well as adjuvant RT,

Adjuvant CT, as well as adjuvant and RT,

Line 284

on multivariate models

on the multivariate models

Line 293-294

from SEER database between 1979 and 2019

from the SEER database between 1979 1975 and 2019

Line 321

after matching procedure

after the matching procedure

Line 324

presented almost triple-negative molecular subtype

presented an almost triple-negative molecular subtype

Line 328

ACC of breast

ACC of the breast

Line 329

5-year and 10-year BCSS rate 95.3% and 92.6%

5-year and 10-year BCSS rates of 95.3% and 92.6%

Line 331

which was a favourable histology

which was a favourable histology

Line 332

Since its rarity

Since Given its rarity

Line 335

ACC of breast

ACC of the breast

Line 336

Among 1,036 included patients from SEER database

Among 1,036 included patients from in the SEER database

Line 339

a negative attitude of physician

a negative attitude of physicians

Line 341

vast majority of patients

the vast majority of patients

Line 342

nodal negative

nodal node negative

Line 346

adjuvant was not necessary

Adjuvant CT was not necessary

Line 350

adjuvant CT could optimize prognosis

adjuvant CT could optimize the prognosis

Line 352

nodal positive

nodal node positive

Line 354-355

younger age, higher histology grade, larger tumor size, nodal involved and ER-negative

younger age, higher histology grade, larger tumor size, nodal involved, and ER-negative

Line 358

failed to improved

failed to improved

Line 361

Among 1,036 patients included from SEER database, 422 underwent adjuvant RT

Among 1,036 patients included from in the SEER database, 422 underwent received adjuvant RT

Line 364-365

The role of adjuvant RT of locoregional control

The role of adjuvant RT of in locoregional control

Line 365

still remained unclear

still remained unclear

Line 370

SEER database

the SEER database

Line 370

the results of univariate stratified analysis

the results of the univariate stratified analysis

Line 374

current guideline recommended

the current guidelines recommended

Line 390

from SEER database

from the SEER database

Line 392-393

The second were limitations in the SEER database, for example, a great loss of Her2 expression and lacks records of other adjuvant treatments such as endocrine therapy.

The second were was limitations in the SEER database, for example, a great loss of missing data on Her2 expression and lacks records of other adjuvant treatments such as endocrine therapy.

Line 394

ACC was a rare histological type

ACC was is a rare histological type

Line 446

https://www.nccn.org/

hHttps://www.nccn.org/
